# Ecological Cutting Fluids

**DOI:** 10.3390/ma13245812

**Published:** 2020-12-19

**Authors:** Marian Włodzimierz Sułek, Anna Bąk-Sowińska, Jacek Przepiórka

**Affiliations:** 1Cardinal Stefan Wyszyński University in Warsaw, 01-815 Warszawa, Poland; 2Dars Cosmetics, 26-600 Radom, Poland; anna.bak@dars.pl; 3Faculty of Chemical Engineering and Commodity Science, Kazimierz Pulaski University of Technology and Humanities in Radom, 26-600 Radom, Poland; jacek.przepiorka@uthrad.pl

**Keywords:** surfactants, metalworking fluids, surfactant working fluids, surfactant lubricants, friction mechanisms

## Abstract

This study deals with metalworking fluids (MWFs). According to DIN 51385, depending on their base, the fluids are divided into oil and water fluids. The oil bases include, among others, mineral, synthetic, vegetable and paraffin oils. This division does not comprise surfactant solutions which can be successfully used in metalworking. Due to the fact that this type of fluid was not qualified, a new type of lubricant based on the aqueous solutions of surfactants has been proposed. Two new notions have been introduced: surfactant working fluids (SWFs) for working fluids and surfactant lubricants (SLs) for all kinds of lubricants as a broader term. The effect of the physicochemical properties of aqueous solutions of sodium lauroyl sarcosinate (SLS), also known as sodium *N*-dodecanoyl-*N*-methyl glycinate, on tribological properties determined using a four-ball machine (Tester T-02) (Łukasiewicz Research Network—The Institute for Sustainable Technologies, Radom, Poland) was analyzed. On the basis of aqueous SLS solutions a composition of working fluids was developed and their functional properties were verified by means of tribological and stand tests as well as during operation. The test results obtained clearly indicate that functional properties of 2% solutions of sodium lauroyl sarcosinate with a foam inhibitor (0.05%) and a biocide (0.1%) are comparable to those of a quality commercial hydraulic fluid.

## 1. Introduction

According to DIN 51385, depending on their base, metalworking fluids are divided into oil and water types [1]. The choice of an MWF base depends on the kind of the production process, the material worked and the geometry of the product and the tool. Oil-based fluids are applied when good lubrication is favoured while water-based fluids are used when it is essential to reduce the temperature of the materials and wear products by heat dissipation outside of the friction zone. Metalworking oils are composed using animal, vegetable or synthetic oils as a base.

Aqueous bases of metalworking fluids can be divided into solutions of chemical compounds and the o/w type emulsions. The solutions of chemical substances, also known as synthetic working fluids, are commercially available in the form of concentrates. They do not contain hydrophobic components. Emulsion working fluids can be categorized as colloids. Their stability is determined by high dispersion of oil as a dispersed phase (micro- and nano-emulsions) and effective emulsifiers. They are available commercially in the form of concentrates. Additives modifying physicochemical, tribological and functional properties are added to all kinds of metalworking fluids.

The analysis of aqueous solutions of surfactants indicates that they have not been included in any of the presented types of working fluids. In view of the possible application of aqueous surfactant solutions as lubricants, which was documented in numerous publications [2,3,4,5,6,7,8,9,10,11,12,13,14,15,16,17,18,19,20,21,22,23,24,25,26,27,28,29,30], it was proposed to create two new groups termed surfactant working fluids (SWFs) and surfactant lubricants (SLs). The aims of the introduction of this new classification (SWFs and SLs) are to avoid ambiguity in the description and to create a basis for a targeted development of this type of working and hydraulic fluids and, in the long term, a different group of lubricants. SWFs and SLs are often mistakenly equated with emulsions or aqueous solutions. There are a number of substantial differences justifying a need to create a novel group of working fluids.

Surfactant lubricants can occur as real solutions, micellar solutions or mesophases, whereas emulsions are colloids. Emulsions contain surface active agents but they are used as emulsifiers of the oil phase in water, and thus they differ substantially from the type of components present in SWFs which play the role of surface-active compounds. Aqueous solutions of chemical compounds contain hydrophilic components which are not surfactants. Additionally, action mechanisms of SWFs, emulsions and aqueous solutions are different under working conditions.

## 2. Aqueous Solutions of Anionic Surfactants as SWF Bases

As an introduction, it seems useful to present basic information about formation and properties of aqueous solutions of anionic surfactants with special attention being paid to the formation of ordered structures at the solid/solution interface. Surfactants are a group of amphiphilic compounds whose common feature is a change in physicochemical properties at the interface. They exhibit relatively high surface activity which is characteristic of the system and depends not only on the kind and properties of surfactants but also on the solvent (water) and other compounds present in the solution. According to the Gibbs theory, the surface phase being in equilibrium with the bulk phase is formed at the solid-surfactant solution interface. The area of the bordering phases is called the surface phase or the interfacial layer (Figure 1). The concentration of surfactants in the surface phase can be many times higher than in the bulk phase. The surface activity of these compounds results from the specific structure of their molecules.

### 2.1. Adsorption at the Interface: Solid-Surfactant Solution

Two parts can be distinguished in the structure of surfactants: hydrophobic which is normally an alkyl chain and hydrophilic (various functional groups). Hydrocarbon chains introduced to water cause breakage of intermolecular hydrogen bonds between water molecules, whose number depends on the length of the chain. This process is thermodynamically unfavorable (∆G > 0). The entropic contribution connected with breakage of hydrogen bonds (hydrophobic effect) has a decisive impact on the free enthalpy increase. Therefore, hydrophobic compounds do not dissolve in water. The specific behavior of amphiphilic molecules, also containing hydrophilic parts, consists in limiting a direct contact of hydrocarbon chains with water. This effect can be achieved by strong adsorption of surfactants at the interface with the formation of structures preventing a direct contact of the hydrophobic parts with water. This is a dominant process and, in the case of molecules with a long alkyl chain, the concentration of surfactants in the surface phase is considerably higher than in the bulk phase. With an increase in concentration, surfactant molecules produce, in turn, monolayers, closed structures (micelles) and then mesophases in the surface phase [31]. All the structures are spatially arranged in such a way that hydrophilic parts are turned outwards so that any contact of the hydrophobic part with water is impossible. When the processes are over in the surface phase, analogous structures are formed in the bulk phase. In view of the interaction with the surface, the geometry of monolayers, micelles and surface mesophases is complicated. It is easier to illustrate the structures formed in the bulk phase. A model structure of a spherical micelle can be used as an example (Figure 2).

With an increase in surfactant concentration, micelles interact with one another and can produce long-range ordered structures called mesophases or lyotropic liquid crystals (LLCs) (Figure 3).

Anionic surfactants are widely used in various branches of industry and their production rate worldwide is significant. Three groups of compounds have been tested from the viewpoint of their application as bases of surfactant working fluids (SWFs).

### 2.2. Alkyl Sulfates and Alkyl Ether Sulfates

Alkyl sulfates and alkyl ether sulfates were represented in this study by sodium dodecyl sulfate (SDS) and sodium laureth sulfate, also known as sodium lauryl ether Sulfate (SLES). In comparison with SDS, SLES contains in its molecular structure two mers of ethylene oxide which affect an increase in hydrophilicity and solubility in water. This results in decreases in conductivity, ionic nature and skin irritation. The physicochemical properties of aqueous solutions of these surfactants indicate the formation of micelles and lyotropic liquid crystals (LLC) both in the bulk phase and in the surface phase [6,7]. A hypothesis has been formulated that surface structures may, under friction conditions, form a stable lubricant film which reduces motion resistance and wear and increases the ability of the system to prevent seizure. This thesis has been confirmed in numerous tribological tests carried out under heavy loads (0–7.2 kN) [6,7]. In the interpretation of the results obtained, an important role was attributed to high adsorption, characteristic of anionic surfactants, and to the formation of liquid crystalline structures containing easy slip planes. The solutions are expected to be used as working fluids and fire-resistant hydraulic fluids in view of high concentration of water (>95%). The main disadvantages of the solutions of the surfactants (SDS and SLES) used were their excessively high foam formation and corrosivity of metals. Foam and corrosion inhibitors were added to alleviate those properties. An alternative research effort was to search for surfactants which would exhibit surface activity without excessive foam formation and metal corrosion.

### 2.3. Sulfosuccinate Derivatives Produced on the Basis of Polyoxyalkylates of 2-Ethylhexyl Alcohol

The representatives of this group of compounds were four sulfosuccinate derivatives produced on the basis of polyoxyalkylates of 2-ethylhexyl alcohol as a result of oxyalkylation with 1 mole of propylene oxide and 3 moles of propylene oxide, 1 mole of propylene oxide and 9 moles of ethylene oxide, 3 moles of propylene oxide and 9 moles of ethylene oxide and 3 moles of ethylene oxide in the presence of a double metal cyanide catalyst (DMC) [28,30]. They were synthesized within the framework of the Project [32]. The compounds with a sterical hindrance exhibit lower foam formation confirmed by foamability tests (the Bikerman method—Bikerman J. J., 1973, Foams, New York, Springer-Verlag, p. 337). Foam did not appear when the concentration of the surfactant in solution was up to 0.1% while in the case of c > 0.1% the foam volume was about 50 mL after 1 min and over 50 mL after 10 min [28,30]. Thus, the criteria for the discussed applications were satisfied. The surfactants tested exhibited high surface activity and also favourable tribological properties qualifying aqueous solutions of the surfactants for the selected applications.

### 2.4. Sodium Sarcosinates

Sodium lauroyl sarcosinate is a representative of anionic surfactants. Based on the literature data it has been found that apart from exhibiting surface activity it may act as a strong anti-corrosive agent [33,34]. The results of physicochemical and tribological studies of aqueous solutions of these surfactants are presented in this paper.

Based on the studies (Chapter 3) and the literature data [2,3,4,5,6,7,8,9,10,11,12,13,14,15,16,17,18,19,20,21,22,23,24,25,26,27,28,29,30] an attempt was made to try to describe the friction mechanism in a system with sarcosinate solutions and a formulation and production technologies of SWF with solutions of these surfactants were developed. Stand and operational tests were carried out on the SWFs obtained.

## 3. Experimental Results

The aim of the planned physicochemical and tribological tests was to confirm the possibility of application of aqueous solutions of sodium lauroyl sarcosinate as bases for surfactant working fluids (SWF).

### 3.1. Physicochemical Properties of Aqueous Solutions of Sodium Lauroyl Sarcosinate as SWF Bases

Sodium lauroyl sarcosinate (SLS), also known as sodium *N*-dodecanoyl-*N*-methyl glycinate, is an anionic surfactant and an amphiphilic compound consisting of a hydrophobic part (12 carbon alkyl chain) and a hydrophilic part (hydrophilic carboxylate). Its molecular mass is about 293 g/mole and its structural formula is given in Figure 4.

The choice of sodium lauroyl sarcosinate as an active component of surfactant working fluids resulted from an analysis of properties of various surfactants from the viewpoint of the planned application (working fluids). The most important properties are:Environmental safety and human health in the workplace,Stability and biodegradability,Biocidal action on microorganisms,High surface activity in a wide interval.

In order to characterize SLS solutions, tests were carried out on stability, surface tension, wettability, viscosity, foamability and corrosion. The solutions with the concentrations of 0.001%, 0.01%, 0.1%, 1%, 2% and 4% were prepared using the gravimetric method. SLS and distilled water were used to prepare the solutions.

#### 3.1.1. Stability

The evaluation comprised the appearance, color, consistency and homogeneity of fluids stored at room temperature (storage test), at a reduced temperature (5 °C) and at an elevated temperature (60 °C) (temperature tests), as well as exposure to centrifugal force (mechanical load tests). The solutions were stable.

#### 3.1.2. Surface Activity

Just like other amphiphilic compounds, sodium lauroyl sarcosinate exhibits high surface activity as a result of which the surfactant is present in the surface phase in a significantly higher concentration than in the bulk phase (Figure 1). The composition and structure of the surface phase have a decisive impact on the lubricity of the system. The other very important property of surfactant solutions, including aqueous SLS solutions, is spontaneous formation of structures at the interface and in solution (micelles, lyotropic liquid crystals—LLCs).

The literature contains practically a full range of physicochemical test results and their interpretation characterizing SLS solutions [35,36,37]. On their basis the following values were determined: CMC (13–14 mM), surface tension at CMC concentration of circa (ca.) 30 mN/m, surface excess of surfactant at CMC (ca. 2.10^−6^ mole/m^2^), minimal area covered by the hydrophilic part (0.68 nm^2^/mole), number of aggregates (ca. 46), free enthalpy of adsorption (ca. 57 kJ/mole), free enthalpy of micellization (ca. 30 kJ/mole) [36]. The literature data are sufficient for the interpretation of the tribological test results. The quoted physicochemical studies were carried out under specific static friction conditions. The actual characteristics cannot be directly referred to tribological tests or stand and operational tests. They are important for the interpretation of the results and for verification of the hypotheses of the mechanisms describing friction processes.

The surface activity of SLS solutions was investigated by measuring surface tension (σ) (Figure 5) and contact angle (θ) (Figure 6) as a function of surfactant concentration (c) expressed in weight percent.

#### 3.1.3. Surface Tension

*Surface tension* (σ) was determined using the ring tear-off method. The method involves measuring the force needed to lift a platinum ring from the surface of a solution tested. The measurements were carried out using a LAUDA TD 1 C tensiometer (LAUDA Scientific GmbH, Lauda-Königshofen, Germany).

The dependence of surface tension as a function of concentration is shown in Figure 5.

In the 0.001–0.1% concentration range one can observe a decrease in the σ value and at c > 0.01% it stabilizes at the level of about 30 mN/m. Surface tension of the solutions decreases considerably in relation to water (Figure 5).

#### 3.1.4. Surface Wettability

Wetting angle (θ) of a steel surface was determined using the sitting drop method. The measuring unit consisted of a microscope, a camera and a computer with an installed digital image acquisition and analysis software MultiScanBase (Computer Scanning-Systems II, Warsaw, Poland). The measurements were carried out at 25 °C. The measured wetting angle (θ) is a measure of wettability. An increase in the value of a wetting angle is equivalent to a decrease in wettability.

Figure 6 shows the dependence of contact angle on surfactant concentration.

An increase in the contact angle means a decrease in wettability and, vice versa, a decrease in the contact angle results in an increase in wettability. The analysis of Figure 6 indicates explicitly that in the case of a low concentration range (0.001–0.1%) there occurs a relatively rapid increase in wettability whose change rate radically decreases after exceeding the 0.1% concentration and at higher concentrations it reaches the values of over 40°.

Summing up the analysis of variations in surface tension and wettability as a function of surfactant concentration it can be stated that sodium lauroyl sarcosinate solutions show high surface activity above the 0.01% concentration. The nature of changes in those quantities as a function of concentration in the 0.01–1% range is characteristic of the end of formation of the surface phase and the start of formation of micellar structures in the bulk phase.

#### 3.1.5. Kinematic Viscosity (ν)

Kinematic viscosity of the solutions was determined using the Ubbelohde capillary viscometer (INKOM INSTRUMENTS, Warsaw, Poland) at ca. 20 °C. It increased to a small degree (ca. 15%) in the concentration range of up to 4%.

#### 3.1.6. Foamability

Mechanical working processes can be unfavorably affected by too abundant foam. It may change physicochemical properties of the fluid in the friction area, reduce the effectiveness of covering of mating surfaces with a lubricant and cause cavitation. Surfactant solutions usually generate excessive foam. In order to reduce it, SWFs used in stand and operational tests will contain foam inhibitors whose concentrations will be determined on the basis of studies of foamability and lubricity of SLS solutions. Another possibility is to use aqueous solutions of surfactants with a sterical hindrance [28,30].

Foamability of the solutions was determined on the basis of the PN-C-04055:1985 standard [38]. It consists in measuring the volume of foam produced during the flow of air through a liquid (190 mL) placed in a measuring cylinder (1000 mL) after 1 (V_1_) min and 10 (V_10_) min after the flow has stopped. The results are given in Figure 7.

Foam volume increases with an increase in concentration and in the case of the highest concentrations (1%, 2%, 4%) it takes the values from the 480–520 cm^3^ interval. Such high foamability requires the addition of foam inhibitors to the solutions.

#### 3.1.7. Corrosion

When selecting a surfactant, particular attention was paid to limiting metal corrosion. Based on the literature data it has been found that sarcosinates may function as corrosion inhibitors in aqueous solutions [33,34]. On metal surfaces they form multilayer structures which are strongly bonded with the metal surface. They passivate metal surfaces and protect them against corrosion. The studies were carried out according to the PN-92 M-55789 standard [39] (Ford test—Table 1). The test consisted in treating cast-iron chips placed on a Petri dish with a fluid. The test lasted for two hours. The test result is given in the form: F3-FORD-TEST method symbol/degree of rusting according to Table 1.

Two out of three 1% SLS solutions tested did not exhibit corrosion (F3/0) and one of them showed traces of corrosion (F3/1). Therefore, tests were carried out for 2% solutions. None of them showed corrosion (F3/0) and those solutions underwent stand and operational tests.

### 3.2. Tribological Properties of Aqueous Solutions

A four-ball machine (Tester T-02) (Łukasiewicz Research Network—The Institute for Sustainable Technologies, Radom, Poland) whose description and experimental methodology were presented in the literature [40] was used in tribological investigations. Two types of tests were conducted: at a constant load and at a linearly increasing load. The arithmetic mean values are given in Figure 8. The error was determined according to Student t-testdistribution for the confidence level of 0.95. The errors were not presented in the diagrams but they were described in the text in order to make the figures easy to read.

#### 3.2.1. Tests at Constant Load (1, 2, 3, 4, 5, 6, 7 kN)

The tests were carried out in the load range of 1 to 7 kN. The coefficient of friction (µ) and wear scar diameter (d) were determined and the results are given in Figure 8.

The dependence of the coefficient of friction as a function of concentration is shown in Figure 8a. The maximum measurement error was 0.01. The analysis of the results (Figure 8a) indicates a significant decrease in motion resistance in the presence of surfactants. The µ values for water are 0.38, 0.42 and 0.44 for the loads of 1.0, 1.5, 2.0 kN, respectively. At 2.5 kN the system undergoes seizure in the presence of water. Whereas all the solutions, also at the lowest SLS concentration, do not undergo seizure up to the load of 3.0 kN. In the case of higher loads (*p* > 3 kN) seizure occurs also for solutions with higher concentrations. The exception are solutions with 2% and 4% concentrations for which the system does not undergo seizure even at the highest load (7 kN). In the case of the highest concentrations (2% and 4%) and high loads (> 3 kN) the values of µ are comparable and range from 0.09 to 0.13 without any visible change tendencies. For those friction conditions it is possible to postulate the formation of a stable lubricant film which does not undergo destruction even at 7 kN. Such fundamental differences between water and the solutions suggest a totally different friction mechanism for the two media. The µ(c) function is characterized by a rapid decrease in the coefficient of friction at lower concentrations and stabilization for c > 0.1%.

Based on Figure 8a, one can notice three concentration intervals with various rates of decrease in friction coefficients as a function of surfactant solutions, particularly for the loads of 1, 2 and 3 kN for which seizure is not observed for the whole range of surfactant concentrations (0.001–4%). The first range (0.001–0.01%) is characterized by small changes, the second range (0.1–1%) shows definitely the highest rate of changes and the third range (0.1–4%) in which the coefficient of friction stabilizes. In the first range the surface layer consists primarily of monomers. Its composition is insufficient to ensure good lubrication and carrying of high loads. In the second range the surface layer is enriched with a surfactant to such an extent that a stable lubricant film consisting of ordered structures can form. In the third range the structure of the formed film is stable and the system can carry high loads at relatively low coefficients of friction (0.11–0.16). Excellent results were obtained at high loads (5–7 kN) and a concentrated contact and the surfactants used can be regarded as effective friction modifiers.

The lack of a significant effect of load on motion resistance for the concentrations of ≥ 1% indicates that the stability of the lubricant film being formed is independent of the loads in the 1–7 kN range. The surfactants display strong adsorption and the surface structures formed do not disintegrate. With an increase in load, the lubricant film is permanently bonded with the friction surface and is not squeezed out of its area.

Figure 8b shows the dependence of wear scar diameter (d) on surfactant concentration (c) in the 1–7 kN load range. The points in the charts are arithmetic means from three independent measurements, each of which is an arithmetic mean from two values of wear scar measured in the perpendicular direction to each other. The measuring error was estimated as 0.1 mm. In the case of the lowest loads (1, 2 kN) for which it is possible to compare wear for water and the solutions, the d values for the solutions are over two times lower, compared to water. Generally, wear scar diameter decreases with an increase in surfactant concentration and increases with an increase in load, which is a predictable tendency. The highest concentration solutions (1%, 2%, 4%) behave differently. For *p* > 4 kN for which wear scar diameter practically does not depend on surfactant concentration and load, the d values range from 1.8–2.0 mm. A decrease in wear scar diameter as a function of increasing surfactant concentration at a constant load and a relatively small effect of load on the d value for surfactant concentrations higher than 1% confirm the earlier thesis that, with an increased additive concentration, adsorptive layers form and successfully reduce the effect of a load increase on wear.

The analysis of the tribological test results at constant loads (Figure 8) points to a significant role of surfactants in reducing motion resistance and wear. The generally observed changes (Figure 8) correspond well with the changes in physicochemical properties (Figure 5 and Figure 6). A significant decrease in surface tension (σ) and an increase in contact angle (θ) are accompanied by a significant reduction in motion resistance and wear. On this basis it is possible to formulate a hypothesis that this results from formation of surface structures which, under friction conditions, are transformed into a lubricant film which decreases motion resistance and wear. It is permanently bonded with the friction area and is not squeezed out of it even at high loads (7 kN).

#### 3.2.2. Tests at Linearly Increasing Load

The tests were carried out on a four-ball machine (Tester T02) at a linearly increasing load at the speed of 409 m/s, in the load range of 0 to 7.2 kN and a spindle speed of 500 rpm. The measurement methodology was presented in the literature [40]. The quantities determined were: scuffing load (P_t_), seizure load (P_oz_), wear scar diameter (d_oz_) and limiting pressure of seizure (p_oz_).

#### 3.2.3. Scuffing Load (P_t_)

Friction torque (M_T_) was a directly measured quantity. A typical course of M_T_(P) is given in Figure 9.

In the load range of 0 to the value of P_t_ the measured friction torque increases relatively to a small degree but there occurs a sharp increase after exceeding the value.

Figure 10 shows the dependence of the P_t_ values as a function of an increasing surfactant concentration. The mean measuring error can be estimated as ca. 100 N.

The P_t_ values determined for individual surfactant concentrations are a measure of lubricant film stability. It follows from Figure 10 that even a threefold increase in the P_t_ value can be observed already at the lowest SDS concentration (0.001%). Scuffing load remains practically constant in the 0.001–0.01% concentration range while in the range of above 0.1% there occurs a sudden, over 2.5-fold increase to the value of 1600 N. Above 3% the P_t_ values are comparable within a margin of error. A correspondence with changes in surface tension and wettability (Figure 5 and Figure 6) as a function of surfactant concentration becomes obvious. High stability of the lubricant film may thus result from self-ordering of surfactant molecules at the interface and formation of surface structures.

#### 3.2.4. Seizure Load (P_oz_)

This is a load at which friction torque has exceeded 10 N.m. The values of P_oz_ as a function of increasing load are given in Figure 11. The mean measuring error can be estimated as 200 N.

The value of P_oz_ is comparable with that determined for water only in the case of the lowest concentration (0.001%). An increase in surfactant solution results in a significant increase in seizure load whose value ca. 7 kN is reached by the solutions with highest concentrations (2.4%).

#### 3.2.5. Wear Scar Diameter (d_oz_)

The measurements of wear scar diameters of the balls were carried out after the test using a POLAR reflection microscope produced by PZO-Warsaw (Poland). Wear scars of the balls were measured perpendicular and parallel to the direction of rubbing and an arithmetic mean of the two measurements was calculated. The values given in Figure 12 are an arithmetic mean of three independent measurements. The error in d_oz_ determination was estimated as 0.1 mm.

Based on the analysis of the results (Figure 12) one can observe an important effect of surfactant concentration on wear at higher surfactant concentrations (c > 0.1%). The measured wear scar diameter decreases even more than twofold which indicates an effective separation of mating surfaces of the friction pair and effective wear reduction.

#### 3.2.6. Limiting Pressure of Seizure p_oz_

Limiting pressure of seizure can be calculated from the equation:(1)$$p_{\mathrm{oz}}=0.52\;\frac{P_{\mathrm{oz}}}{d^{2}}$$

The p_oz_(c) dependence is presented in Figure 13.

The value of p_oz_ for the solution with the highest surfactant concentration is 11-fold higher in relation to water.

Aqueous solutions of sodium lauroyl sarcosinate successfully protect the system against seizure. At the lowest concentrations of surfactant solutions (0.001–0.01%) the quantities characterizing wear resistance have more favourable values than those for water and the differences between the two media increase with an increase in SLS concentration. In the case of solutions with the highest concentrations (3%, 4% and 5%) hydrodynamic friction occurs at a relatively high load (ca. 1500 N). The system undergoes seizure at about 7100 N which is comparable with the maximum value allowable for this type of tribometer (7200 N). At 7100 N wear scar diameter takes the value of 1.3 mm which is more than two times lower than the one for water at a significantly lower load (2 kN). High P_oz_ values and low d_oz_ values result in a high value of limiting pressure of seizure (p_oz_) which is even about 11-fold higher than the one for water. In the case of the surfactant solutions studied, it is interesting to note a characteristic correlation of the tribological test results (Figure 10, Figure 11, Figure 12 and Figure 13) with the physicochemical test results (Figure 5 and Figure 6) and this can also be observed in constant load tests.

## 4. Surfactant Working Fluids (SWFs) Friction Mechanisms

Aqueous surfactant solutions used as lubricants (SWFs) play an increasingly important role in fundamental and applied research [2,3,4,5,6,7,8,9,10,11,12,13,14,15,16,17,18,19,20,21,22,23,24,25,26,27,28,29,30]. They are environmentally friendly and safe for human health in the workplace [37,41,42]. The main current SWF application trends are operational (working and hydraulic) fluids. The interest in this type of lubricants results from the analysis of friction processes in biological systems. They inspire and set directions for engineering applications.

The Zhu model [43] is the simplest adsorption model which can be applied to describe the formation of the surface phase. It assumes that at the lowest surfactant concentrations monomers adsorb on the surface and they successively occupy free sites. In this region (region I) the adsorption isotherm increases to a small degree and the surface becomes successively filled with monomers. After reaching the critical surface aggregate concentration (CSAC) the excess function increases intensively and surface micelles form on the surface and then mesophases form at higher concentrations. The formation of long-range ordered surface structures ends after reaching the CMC values at which micelles begin to form in the bulk phase. For c > CMC the value of surface excess remains constant (plateau). The extension of the Zhu model is a four-region model which divides the isotherm into four regions differing in its slope [31]. A two-region model is sufficient for interpretation of the results.

The formation of structures in the surface phase is described on the basis of the results of tests carried out on surfaces with defined composition, structure and geometry by means of modern measurement techniques. Therefore, a quantitative transfer of these test results to engineering practice and/or tribological studies is burdened with a big error and pointless. However, on the basis of this research it can be assumed that at the surfactant solution-solid surface interface there forms an interfacial region with a significantly higher surfactant concentration in comparison with the volume of the solution. Self-ordering structures (micelles and mesophases) are produced there. The physicochemical properties, structure and composition of the two phases differ significantly. Based on the test results presented (Figure 5, Figure 6, Figure 8, Figure 10, Figure 11, Figure 12 and Figure 13) it can be definitely stated that formation of the surface phase was ended at the highest surfactant concentrations (c > 0.1%). Under friction conditions the surface phase is transformed into a lubricant film which separates mating surfaces, reduces motion resistance and wear and improves the ability of the system to prevent seizure.

A probable friction mechanism for anionic surfactant solutions was presented on the basis of the literature and our own [2,3,4,5,6,7,8,9,10,11,12,13,14,15,16,17,18,19,20,21,22,23,24,25,26,27,28,29,30] data. It assumes that the mechanism of lubrication in oil bases and aqueous surfactant solutions is definitely different. Oils are commonly used for various friction pair materials in order to reduce friction and wear. They perform their functions even at high loads due to a favourable dependence of viscosity on temperature and pressure. However, the application of oil-based lubricants has a negative effect on natural environment and human health [41].

Water-based lubricants are an alternative for oils. However, it should be pointed out that both emulsions and aqueous solutions of compounds may be dangerous, particularly used emulsions which are treated as hazardous waste. Only surfactant working fluids (SWFs) can be regarded as a new generation of lubricants which satisfy the criteria of the so-called green chemistry. In spite of their unquestionable ecological advantages, common application of SWFs in lubrication technology requires numerous studies and introduction of bold, innovative solutions.

Petroleum-based products consist of unassociated molecules which, on friction surface, produce a lubricant film consisting of about 5–8 molecular layers. With an increase in pressure and shortening the distance between friction pair surfaces, the thickness of oil film decreases and its density increases. At relatively high pressures, oil reaches the yield point and after exceeding the point it takes the properties of a solid [44]. In the case of water, an increase in density and a transfer to a solid under pressure are not possible. The density of ice is lower than the density of water. For this reason water has insufficient lubricity. Water molecules do not produce a lubricant film and are removed out of the friction area which results in the contact of micro-asperities and, subsequently, in an increase in motion resistance and wear and in scuffing. The situation changes dramatically when an anionic surfactant is added to water and it dissociates into cations (sodium ions) and anions (hydrophilic part of the surfactant). Due to a relatively small radius, sodium ions are strongly solvated and the exchange of water molecules with the hydration sheath occurs with high frequency (10^9^ s^−1^) [44]. As the studies carried out for solutions of simple salts (NaCl, KCl) have shown, the solvation sheaths of Na^+^ and K^+^ ions are particularly stable and hard to remove out of the friction area. Their coefficients of friction are low [44,45]. They can be treated as “molecular bearings” located in micro-asperites.

The research quoted was done for model systems (mica surfaces, aqueous solutions of NaCl and KCl) [44,45]. The results obtained were interpreted as an effect of stability of a lubricant film consisting of hydrated ions with slip planes inside. These conclusions can be applied also to solutions of anionic surfactants. Their characteristic feature is high surface activity due to which the concentration of sodium ions in the surface phase is so high that hydrated, strongly repelled sodium ions produce a lubricant film with a high load-carrying capacity and low motion resistance and wear (Figure 8, Figure 10, Figure 11, Figure 12 and Figure 13).

The hydration of hydrophilic parts of sarcosinates (anions) and within mesophases also affects lubricity. Layers of water solubilized inside liquid crystalline structures are formed due to interaction of water dipoles with a negatively charged hydrophilic part. They give rise to an increase in stability of a liquid crystal and produce an additional easy slip plane inside the layers. In order to illustrate this phenomenon one can give an example (probably not perfect) of skates sliding on ice. In this case water produced due to ice melting is the slip layer.

The effect of the hydrophobic part of a surfactant on tribological properties is more complex. It can generally be said that the alkyl chain determines the composition and structure of the surface phase. Its length and spatial structure have an effect on the degree of surfactant packing, types and sizes of micelles and mesophases in the surface phase, as well as on solubilization of water and other components in micelles and lyotropic liquid crystals.

On the basis of this analysis it can be said that physicochemical and tribological properties of water and aqueous solutions are totally different. It turns out that tribological properties of the solutions can be correlated with the properties which characterize adsorption at the solid-solution interface. In this paper surface tension (σ) and wettability (θ) were a measure of surface activity of the solutions tested. The 0–0.1% concentration range at which a considerable decrease in surface tension and an increase in wettability occur is characteristic (Figure 5 and Figure 6). These values correspond well with the tendencies for variations in tribological properties as a function of concentration (Figure 8 and Figure 10, Figure 11, Figure 12 and Figure 13). These correlations can be interpreted in terms of formation of the surface phase resulting from adsorption of the surfactants and hydration of its ions (cations and anions), micellar and LLC solubilization and specific properties of the surface phase.

An analysis of the friction mechanism of SWFs should include the structure and physicochemical properties of surfactants, particularly their high surface activity, formation of surface structures, solvation of sodium ions and anions, hydration of micelles and mesophases. Summing up, it can be stated that a stable lubricant film forms under friction conditions and it is not squeezed out of the friction area and exhibits high load carrying capacity.

## 5. Stand and Operational Tests

As a result of the physicochemical and tribological studies (Figure 5, Figure 6, Figure 7 and Figure 8 and Figure 10, Figure 11, Figure 12 and Figure 13), a formulation of a stable, non-foaming working fluid (SWF) was developed. It did not cause corrosion and its viscosity was comparable with that of water. It underwent physicochemical and tribological tests, as well as stand and operational tests. The results were related to a quality commercial fluid.

### 5.1. The Composition of a Surfactant Working Fluid (SWF)

Water with hardness of 10°n (in German degrees) prepared according to the procedure described in the PN-92/M 55789 standard. The hardness of this kind of water is close to that of commonly used tap water in Poland,Sodium lauroyl sarcosinate (2% concentration),Antifoam additive (Silfoam SE 47-silicone emulsion, 0.05% concentration),Biocide (Euxyl K120—mixture of methylchloroisothiazolinone and methylisothiazolinone—0.1% concentration).

Corrosion inhibitors were not used due to the absence of metal corrosion in 2% solutions of sodium lauroyl sarcosinate.

### 5.2. Physicochemical and Tribological Properties

Physicochemical and tribological tests were carried out for SWFs with the given composition and their results are shown in Table 2 and Table 3.

### 5.3. Test Stand Studies

The tests were carried out at the Institute of Advanced Manufacturing Technologies in Cracow (Kraków, Poland) for the developed surfactant working fluid and a commercial fluid as a reference.

Two independent kinds of tests were conducted on:Tool life of sintered carbide cutting edges at straight turning of standard steel C45.Tool life of a cutting edge and cutting resistance using the WBS device.

### 5.4. Tool Life of Sintered Carbide Cutting Edges at Straight Turning of Standard Steel C45

Tool life of sintered carbide blades at straight turning of standard steel C45 was studied on a machine tool (DMG MORI, Tokyo, Japan) (lathe-miller center NL 2000S4 Mori Seiki-driving engine power of 18.5 kW). The worked material was higher quality standardized construction carbon steel C45 according to PN-EN 10083-8: 2008 [46] in the form of rollers with initial diameter of ca. Φ 180 mm, 600 mm long and with hardness of 167–179 HB. The following cutting parameters were used during turning: cutting speed v_c_ = 180 m/min, feed f = 0.16 mm/r, depth of cut α_p_ = 1.6 mm. In the presence of the working fluid tested, the average tool life of the cutting edge equaled 16.5 min. while for the reference fluid it was 17.5 min. The mean values were calculated from three independent measurements. The comparisons of the values of the average results show no significant differences. Surface roughness whose measure were average R_a_ values was as follows:In the case of turning using cutting edges in the initial phase of wear (VB up to 0.05 mm) the R_a_ values were 1.8 µm for the original fluid and 1.5 μm for the reference fluid,In the case of turning with cutting edges with a new degree of wear (0.05 < VB ≤ 0.30 mm) the R_a_ values were 2.3 μm for the original fluid and 2.0 μm for the reference fluid.

At the depth of cut of α_p_ = 1.6 mm, chips were obtained in the loose form both for the original and the reference fluid (chip shape classification PN–ISO 3685) [47] (Figure 14).

Operational tests were carried out in the P.P.H. RADMOT Jan Stańczyk Company Radom, Poland and in Radom Arms Factory LLC.

In the former Company the fluids were tested on an automatic lathe used for machining of bars SPRINT 65 Linear (GILDEMEISTER Drehmaschinen, Bielefeld, Germany). It is used to produce very complex elements which are made under production conditions in a continuous way. Using the original fluid it was possible to achieve average life of internal tools equal to 250 elements and average life of external tools of 264 items. In the case of the commercial fluid the same values of average life were obtained for both internal and external tools—224 items.

During working with the original fluid it was possible to obtain, on tool surfaces, the mean values of the surface roughness parameter R_a_ which are comparable and equal R_a_ = 1.9 µm for internal tools and *R_a_* = 2.0 µm for external tools. In the case of the commercial fluid, the obtained mean values of the surface roughness parameter R_a_ were R_a_ = 2.0 µm for internal tools and R_a_ = 1.7 µm for external tools.

Operational tests conducted in Radom Arms Factory LLC–(Original name: Fabryka Broni “Łucznik”—Radom Sp. z o.o.) consisted in carrying out tests on 4-axis machining centres BROTHER (BROTHER INDUSTRIES, LTD., Nagoya, Japan) and AVIA (Precision Machine Tools Factory “AVIA”, Warsaw, Poland) and on a CNC lathe WAGNER (Wagner Machine Co. Champaign, IL, USA) during regular production of small arms parts.

Based on the tests it was found that:All fluids displayed good cooling and lubricating properties. No significant differences were observed compared to commercial fluids,No excessive tool wear or deterioration of quality of the worked surfaces were observed,No corrosion occurred on surfaces of the elements worked or machine parts.

Based on the stand and operational tests as well as physicochemical tests it can be stated that the developed working fluids meet the required evaluation criteria allowing for the anticipated applications.

## 6. Conclusions

Numerous research results indicate that aqueous solutions of surfactants play a decisive role in lubrication of biological systems. The analysis of these processes is an inspiration for implementation of those results in materials solutions in engineering systems in which it is required to ensure complete safety of humans and the natural environment at three stages of life: production, operation and disposal. From among the surfactants available on the market it is possible to choose compounds whose safety guarantee are their numerous applications in medicine, cosmetics and household products [37,41,42,48].

This paper presents the results of physicochemical and tribological tests on aqueous solutions of sodium lauroyl sarcosinate which exhibit low motion resistance and wear, as well as a high antiscuffing capacity. The choice of sarcosinate was important because, apart from its high surface activity, it effectively protects steel surfaces against corrosion. SLS can thus be regarded as a multifunctional additive. High load values (ca. 7 kN) were obtained both in tests at a constant load and at a variable load at which the system did not undergo seizure. The values are comparable with maximum allowable loads for this type of tribometer (Test T02).

Changes in tribological properties as a function of surfactant concentration (c) correspond well with variations in surface tension and wettability as a function of c. The comparison of the results obtained and the CMC values for aqueous sarcosinate solutions [37] indicate explicitly that favourable tribological properties are a result of micelles and mesophases generated at the interface. In particular, long-range ordered structures (LLCs) can be important for high load carrying. In the case of anionic surfactant solutions, hydrated ions may have an additional and essential impact on lubricant film stability.

Just like solutions of other surfactants aqueous sarcosinate solutions they can be used in lubrication technology [2,3,4,5,6,7,8,9,10,11,12,13,14]. In order to confirm this thesis, we planned to carry out stand and operational tests for working fluids (SWF) composed of aqueous solution of sodium lauroyl sarcosinate (2%), antifoam additive (0.05%) and biocide (0.1%). The results of test stand and operational measurements of a surfactant fluid (2% SLS solution) were compared with those for a high-quality commercial fluid. The physicochemical and tribological properties at a constant load (2 kN) were comparable for both fluids. However, the surfactant fluid displayed more favorable antiseizure properties (test at an increasing load). The analysis of the test stand and operational investigations carried out in two industrial plants specializing in precision engineering did not show any significant differences between the two working fluids studied.

The demonstrated possibility of application of aqueous solutions of surfactants as working fluids is one more confirmation of the need to introduce a new kind of operational fluid (SWF) which has not been included in the current classification [1] and have not been proposed in the literature.

## Figures and Tables

**Figure 1 materials-13-05812-f001:**
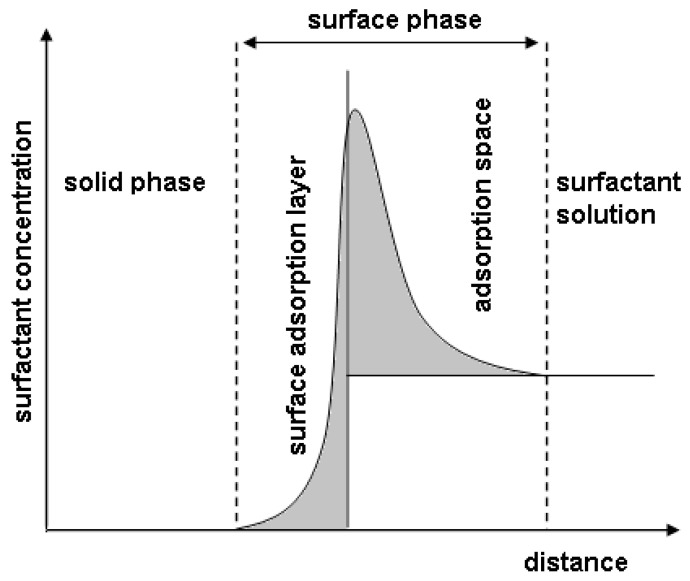
Adsorption at the interface: solid-surfactant solution.

**Figure 2 materials-13-05812-f002:**
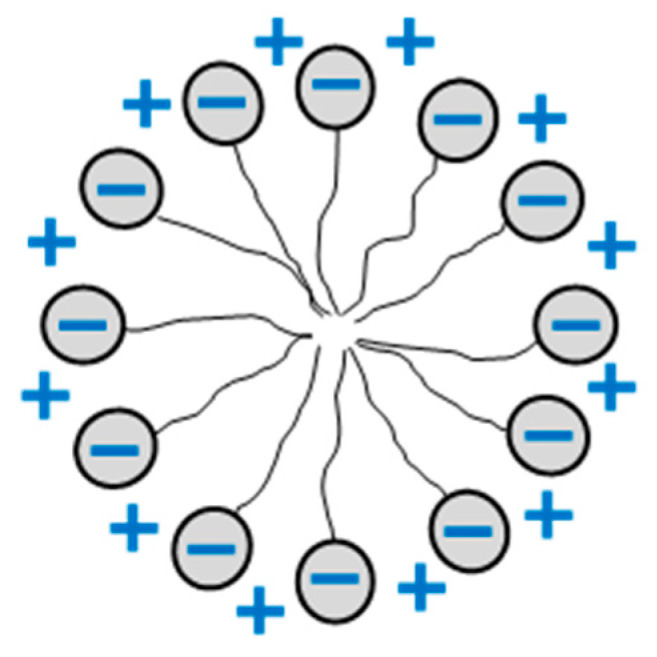
Schematic illustration of an anionic surfactant micelle in solution.

**Figure 3 materials-13-05812-f003:**
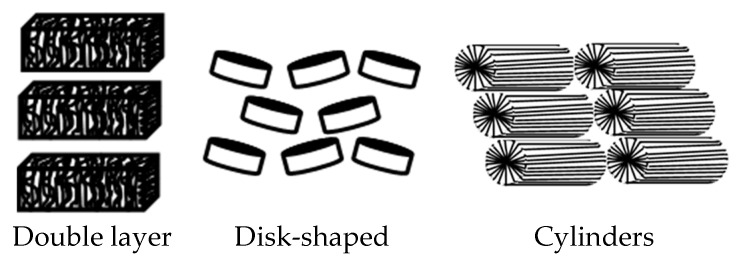
Examples of geometrical models of mesophases formed in surfactant solutions.

**Figure 4 materials-13-05812-f004:**
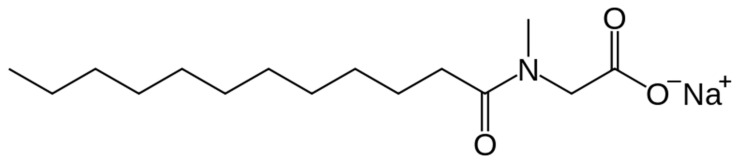
Structural formula of sodium lauroyl sarcosinate.

**Figure 5 materials-13-05812-f005:**
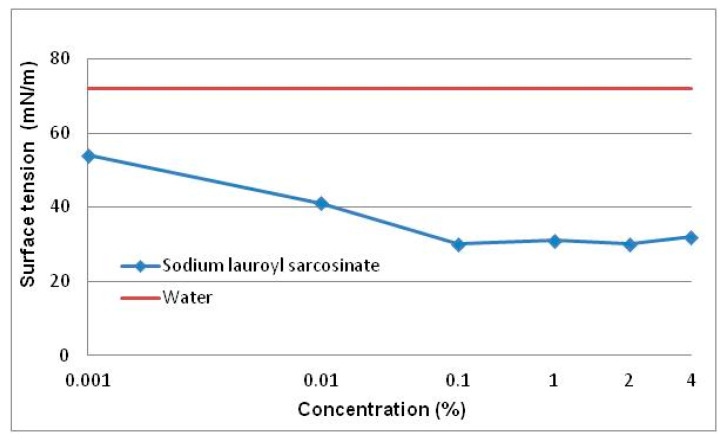
Surface tension of aqueous solutions of sodium lauroyl sarcosinate as a function of its concentration. Measurement temperature ca. 20 °C, Lauda tensiometer. Solid line describes the σ value for water (72 mN/m).

**Figure 6 materials-13-05812-f006:**
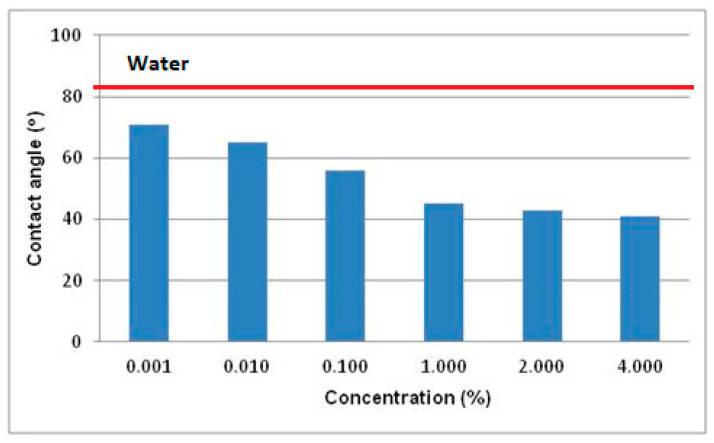
Dependence of contact angle (θ) of a steel surface wetted with aqueous solutions of sodium lauroyl sarcosinate as a function of surfactant concentration (c). Measurement temperature 20 °C, bearing steel (ŁH15—100Cr6—1.3505—high-carbon chromium bearing steel). Solid line describes the θ value for water (81°).

**Figure 7 materials-13-05812-f007:**
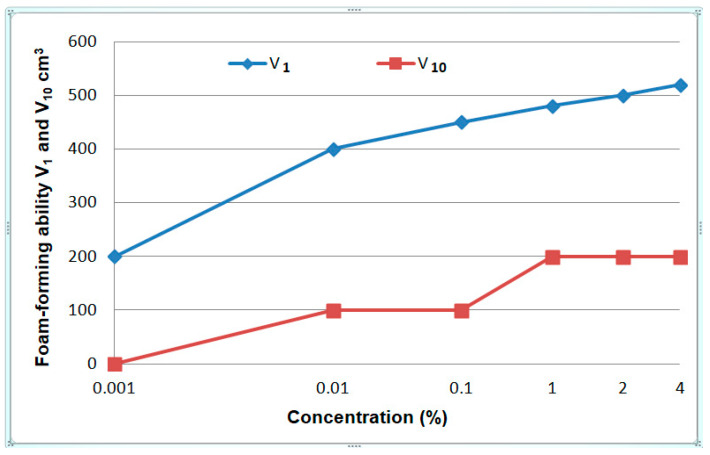
Dependence of foam volume produced as a result of air flow through sodium lauroyl sarcosinate solution after 1 and 10 min after its formation.

**Figure 8 materials-13-05812-f008:**
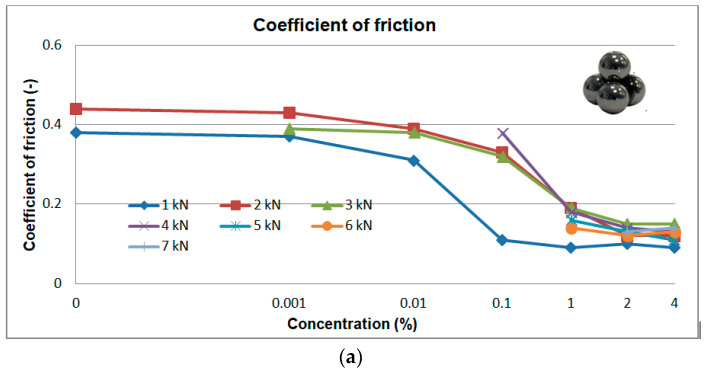
Dependence of coefficient of friction (µ) (**a**) and wear scar diameter (d) (**b**) as a function of load for water and aqueous solutions of sodium lauroyl sarcosinate (SLS). Tribological tester T02, spindle speed of 200 rpm, test duration 900 s.

**Figure 9 materials-13-05812-f009:**
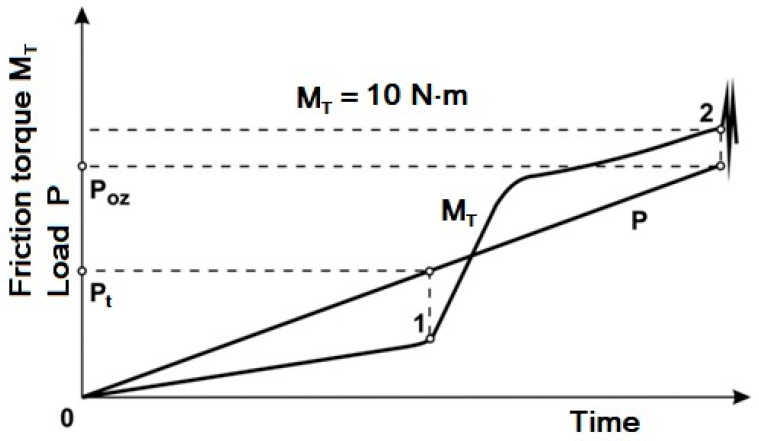
Typical course of changes in friction torque (M_T_) as a function of load (P). Four-ball machine-Tester T-02.

**Figure 10 materials-13-05812-f010:**
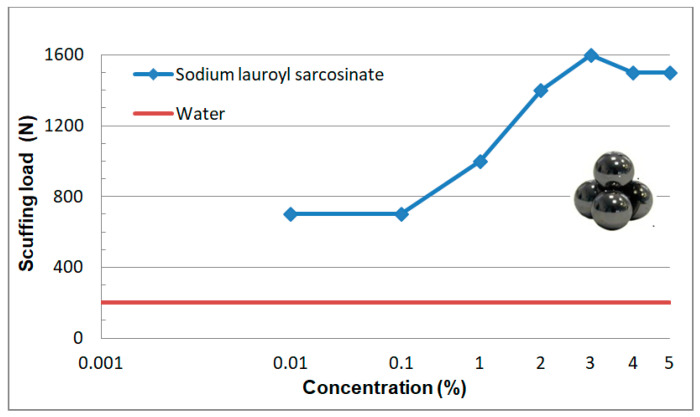
Dependence of scuffing load (P_t_) on concentration of sodium lauroyl sarcosinate (SLS) in aqueous solution. Solid line describes P_t_ value for water (200 N). T02 tester, load increase velocity 409 m/s, spindle speed 500 rpm.

**Figure 11 materials-13-05812-f011:**
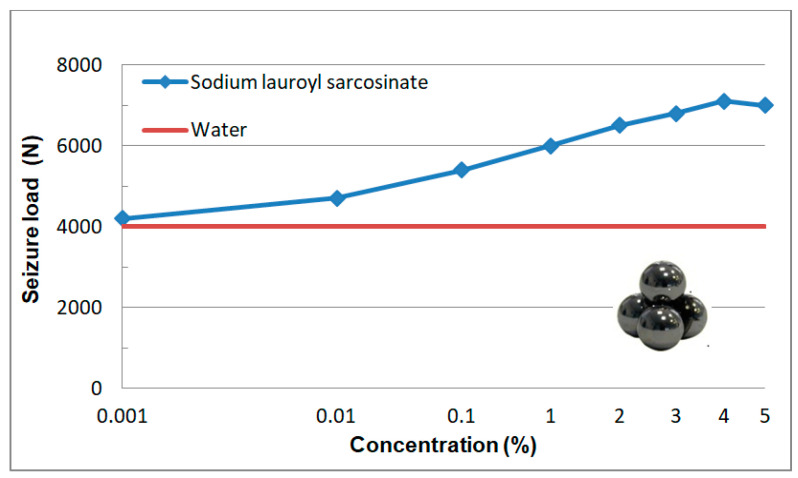
Dependence of seizure load as a function of SLS concentration in aqueous solution. Solid line describes P_oz_ values for water (4000 N).

**Figure 12 materials-13-05812-f012:**
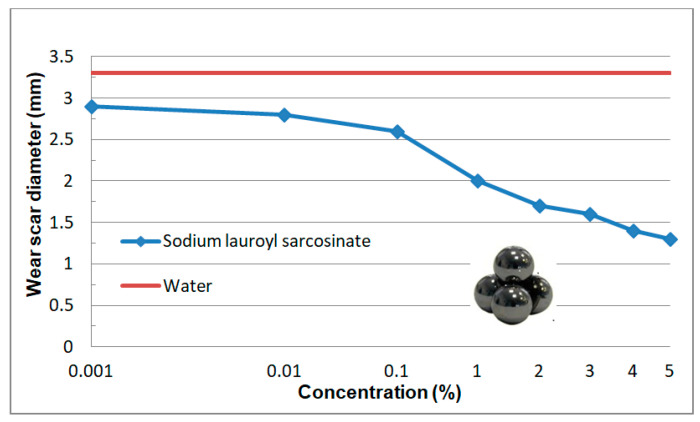
Dependence of wear scar diameter (d_oz_) as a function of concentration of sodium lauroyl sarcosinate. Solid line describes d_oz_ value for water (3.3 mm).

**Figure 13 materials-13-05812-f013:**
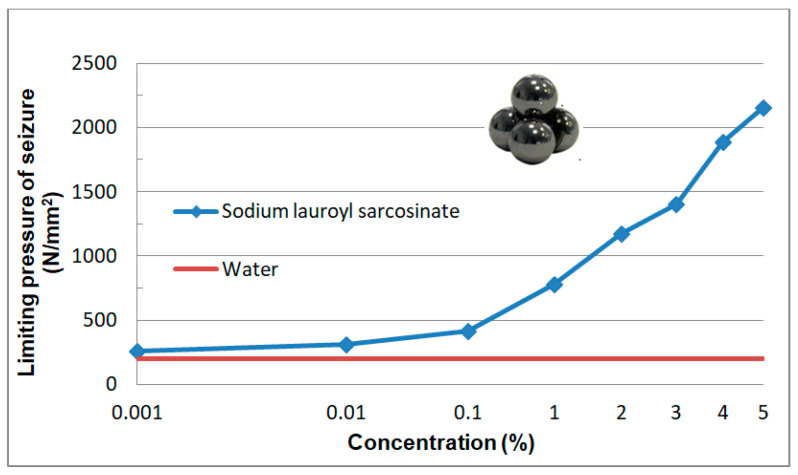
Dependence of limiting pressure of seizure p_oz_ as a function of surfactant solution (c) in aqueous solution. Solid line describes p_oz_ value for water (200 N/mm^2^).

**Figure 14 materials-13-05812-f014:**
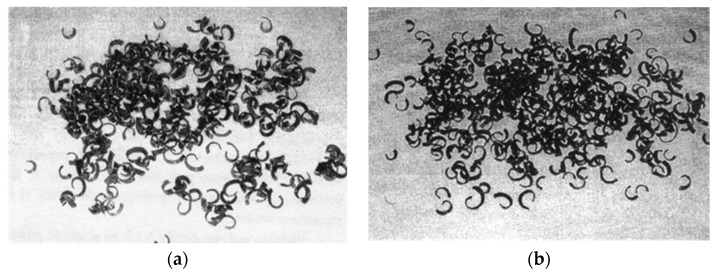
Examples of chips from straight turning using original fluid (**a**) and reference (commercial) fluid (**b**).

**Table 1 materials-13-05812-t001:** Point evaluation of rusting degree according to PN-92/M-55789 standard.

Degree of Rusting	Description	Stained Area (%)
0	no corrosion	no traces of corrosion
1	traces of corrosion	maximum 3 rust spots of up to 1 mm in diameter
2	light corrosion	not more than 1%, but spots larger than for 1
3	moderate corrosion	above 1% up to 5%
4	significant corrosion	above 5% up to 20%
5	strong corrosion	above 20% up to 50%
6	very strong corrosion	above 50%

**Table 2 materials-13-05812-t002:** Physicochemical properties of surfactant working fluids (SWFs) intended for stand and operational tests.

Kind of Fluid	Physicochemical Properties					
	б (mN/m)	θ (deg)	ν (mm^2^/s)	Foamability (mL)	pH (-)	Corrosivity
Water 10°n	72.4	81	1.01	0	7.4	F3/6
Commercial fluid	35	42	0.99	0	9.6	F3/0
2% SLS solutions with additives	31	45	1.10	0	10	F3/0

**Table 3 materials-13-05812-t003:** Tribological properties of selected working fluids intended for stand and operational tests.

Kind of Fluid	Tests at Constant Load T-02—2 kN		Tests at Increasing Load—T-02			
	μ (-)	d (mm)	P_t_ (N)	P_oz_ (N)	p_oz_ (N/mm^2^)	d_oz_ (mm)
Water	0.47	1.8	200	4000	200	3.3
Commercial fluid	0.12	1.1	1000	3900	300	2.6
2% SLS solutions with additives	0.13	1.0	1300	6400	1200	1.7

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
