# Peer review of "Ecological Cutting Fluids"

_materials, 2020, doi:10.3390/ma13245812_

Round 1

Reviewer 1 Report

The present paper reports the Ecological Cutting Fluids. To make this paper publishable the authors need to consider following comments: 

- Title: The title is “too short” and include “wide field”. Is it really cover all these fields? Mostly this kind of short/wide titles are related to reviews or chapters.

- Abstract: the manuscript is not written clearly and professionally. If you use abbreviated statement like o/w, you need to mention before that it’s oil/water (line 14). DIN 51385 is a standard and no need to refer it in Abstract; possible to do in manuscript. As well as about Ref 2, it’s like an advertisement/continuation of this previous paper/work.

- Introduction: In line 51, you mentioned more than 25 article references in one phrase! It’s first time I’ve seen it. Also, Ref 30 is in section 3, but Refs 31-33 in introduction. Last paragraph of introduction is “your approach in present paper” regarding the previous works and “improvements” and “novelties”; I don’t see it at all.

- Section 2 (from line 64) name is not so common. Mostly name it “materials and methods” and then in subsections you explain the processes. By the way, materials and suppliers also is unknown. Figures 2 and 3 are simple schematics that easily can be modeled in PowerPoint, Paint, etc. but has so low quality.

- Section 3 should be only results and discussion. For example, what this “3.1.1. Stability” discuss about? It’s not for here.

- Conclusion: you can’t use references in this section! Abstract and conclusion is written in very basic level and not qualified in present form. Here “This paper presents … [3-15]” what that means?! The references show it or your paper? in another sentence “A comparison of the results of the tests of an original hydraulic fluid … gave very good results”, where is the improvement percentage or comparative parameters. It’s not a factory report, it’s a scientific article. First paragraph is not related to conclusion and somehow second paragraph is a start.

- I don’t see scientific report qualification here and authors need to improve it generally. Please follow the instruction of journal and also English skill need to be changed majorly. 

Author Response

Dear Reviewer

We are replying to your remarks and suggestions presented in your Review.

The title of the article

The title is concise and comprises a wide thematic range. This is a conscious decision resulting from the following reasons:

  • Aqueous surfactant solutions are a specific new class of working fluids which should be presented in a wide range.
  • There is a need to raise interest of a wider group of researchers studying working fluids.
  • An expected effect of the increase in the interest in SWFs will lead to an intensive increase in research on this subject and to practical applications of research results.
  • The paper presents the results of physicochemical, tribological, test stand and operational tests. This is a full range of tests which certify a metalworking fluid as fit for use on an industrial scale.  The proposed sequence of tests refers to all working fluids. The article thus covers the issue of working fluids.
  • For the above reasons we propose leaving the title of the article unchanged.

Abstract

  • Following the Reviewers suggestion we have omitted references in the Abstract and shortened the basic information.

Introduction

The large number of references results from the importance of the issue – introduction of a novel group of working fluids to engineering practice and, in future, to standards. The concept of SWFs is based on:

  • a large number of our own studies involving properties of aqueous surfactant solutions and their applications in lubrication technology (patents),
  • numerous publications from around the world which report data on lubricating properties of aqueous surfactant solutions,
  • a relatively large number of reports on applications of aqueous surfactant solutions in lubrication technology.

Chapter 2.

  • Chapter 2 is not a part of the experimental section and does not contain investigation results.
  • The structure and the position of Chapter 2 in this article diverge slightly from the traditional division which results from the specificity of this work.
  • Omitting this chapter would create a gap in the description of aqueous surfactant solutions as this information is used in the following chapters.
  • Figures 2 and 3 will be corrected according to the Reviewer’s suggestion.

Chapter 3.

  • Chapter 3 presents results of experimental studies.
  • Subchapters 2.1 and 3.1. was divided by the Editor into further sections. We intend to ask the Editor to return to the original version without the division. However, the individual elements have been written using a special font. This also applies to stability.
  • In the structure of Chapter 3 there is no clear division into materials and methods relating to all experimental studies.
  • The article describes physicochemical, tribological, test stand and operational tests which differ both in the testing devices used and in methodology. Therefore, the descriptions of the methods were assigned to individual types of investigations. The descriptions were often reduced to a minimum or there were references to literature.
  • The manuscript was supplemented with a brief description of surface tension measurement.

Summary

  • Following the Reviewer’s suggestions, Summary has been expanded and the supplemented material included in the new version of the article.
  • The suggestion concerning the order of paragraphs has also been taken into account.
  • The references to our own literature reports [ 3 – 15] (currently [  ]) have been justified in earlier replies.
  • The results of physicochemical, tribological and test stand studies can be interpreted. In the operational tests carried out in two industrial plants the criteria for the evaluation of the fluids were unambiguous: a new working fluid should not have worse operational parameters than the one currently used in production processes. Hence, the statement about good results in relation to commercial working fluids.
  • The article has been re-checked from the viewpoint of terminology and style by an experienced specialist who co-authored a book on chemical, physical and biological aspects of tribology published by Elsevier.

Following the Reviewer’s suggestion, we have conducted another analysis of the whole article. The division of subchapter 3.1 has been modified and clearly marked. The Abstract has been shortened, the Summary has been supplemented and all the necessary corrections have been introduced.      

Reviewer 2 Report

  1. The novelty of this research should be highlighted in the abstract.
  1. Too much unnecessary introduction content in your abstract. Make your abstract concise.
  2. Typically,abstract should be your thought rather than others. Therefore, it is unusual to use citations in your abstract.
  3. Where are your Method parts? Before your experimental results, you should have a separate part to describe your experiment procedure.
  4. How you measured your surface tension? What kind of equipment you applied in your research. Give name, brand, and parameters.
  5. Figure 14. I can not tell any difference between a and b. Please describe why you put those figures in your article and what is the difference. Also, add a scale bar.
  6. You need to re-construct your article. Make sure every part are in good order.

Author Response

Dear Reviewer,

We are replying to your comments and suggestions following their order in your Review.

Pts 1, 2 and 3.

The Abstract has been changed according to your suggestions.

  • The less important information was deleted.
  • The references have been removed.
  • The need for a new classification of working fluids was justified by introducing a novel kind of surfactant working fluids. This suggestion has never before been presented in literature. 
  • The range of studies carried out on sodium lauroyl sarcosinate solutions in the context of their applications as SWFs has been presented.

Pt 4.

The article presents the results of physicochemical, tribological, test stand and operational tests. This is a full range of tests which certify a metalworking fluid as fit for use on an  industrial scale. The research procedures were assigned to individual types of tests. In our view such a structure improves the clarity of the article.

Pt 5.

The article has been supplemented with a description of surface tension and wetting angle measurements.

Pt 6.

  • One of the criteria of the quality of metalworking on CNC machines are the chip’s shape and size. Long and tangled chips are a sign of improper metalworking, because they hinder their removal from the working zone. Figure 14 shows a proper chip obtained while working with a SWF developed by the authors of this article and one obtained with a commercial fluid. The figures are shown for comparison only and therefore we did not include a band scale.  
  • Before operational tests (Tables 2 and 3) we presented the results of physicochemical and tribological tests for water, commercial fluid and SWF (2% SLS solution with additives). On their basis one can explicitly state that physicochemical and tribological properties at a constant load (2 kN) are comparable and the antiseizure properties of SWF are clearly more favourable. 

Pt 7.

Following the Reviewer’s suggestions the whole article was re-analyzed. The division of subchapter 3.1 was modified and its components were italicized. The Abstract was shortened, the Summary was supplemented and all the necessary corrections were introduced.

Round 2

Reviewer 1 Report

The article can be considered for publication by the approve of the editor. Thanks.

Author Response

Thank you very much for your review.

Reviewer 2 Report

The authors have satisfactorily responded to all my questions. There is only one thing I recommend to change.

Your title is too broad, covering wide scope of subjects. I recommend you select a better title.

Author Response

Dear Reviewer,

Thank you very much for your review.

The title of the article

The title is concise and comprises a wide thematic range. This is a conscious decision resulting from the following reasons:

  • Aqueous surfactant solutions are a specific new class of working fluids which should be presented in a wide range.
  • There is a need to raise interest of a wider group of researchers studying working fluids.
  • An expected effect of the increase in the interest in SWFs will lead to an intensive increase in research on this subject and to practical applications of research results.
  • The paper presents the results of physicochemical, tribological, test stand and operational tests. This is a full range of tests which certify a metalworking fluid as fit for use on an industrial scale.  The proposed sequence of tests refers to all working fluids. The article thus covers the issue of working fluids.
  • For the above reasons we propose leaving the title of the article unchanged.